# LC/MS Analysis of Saponin Fraction from the Leaves of *Elaeagnus rhamnoides* (L.) A. Nelson and Its Biological Properties in Different In Vitro Models

**DOI:** 10.3390/molecules25133004

**Published:** 2020-06-30

**Authors:** Jerzy Żuchowski, Bartosz Skalski, Michał Juszczak, Katarzyna Woźniak, Anna Stochmal, Beata Olas

**Affiliations:** 1Department of Biochemistry and Crop Quality, Institute of Soil Science and Plant Cultivation, State Research Institute, 24-100 Puławy, Poland; jzuchowski@iung.pulawy.pl (J.Ż); asf@iung.pulawy.pl (A.S.); 2Department of General Biochemistry, Faculty of Biology and Environmental Protection, University of Lodz, 90-236 Łódź, Poland; bartosz.skalski@biol.uni.lodz.pl; 3Department of Molecular Genetics, Faculty of Biology and Environmental Protection, University of Lodz, 90-236 Łódź, Poland; michal.juszczak@biol.uni.lodz.pl (M.J.); katarzyna.wozniak@biol.uni.lodz.pl (K.W.)

**Keywords:** sea buckthorn, saponin, LC-MS/MS, oxidative stress, hemostasis, anticoagulant activity, DNA damage

## Abstract

This study focuses on saponin fraction from sea buckthorn (*Elaeagnus rhamnoides* (L.) A. Nelson) leaves. It has known that for example teas from sea buckthorn leaves have anti-obesity properties. The objective of our present experiments was to investigate both the chemical composition of saponin fraction, as well as their biological properties in different in vitro models (using human plasma, blood platelets, and peripheral blood mononuclear cells (PBMCs)). We observed that saponin fraction reduces plasma lipid peroxidation and protein carbonylation induced by H_2_O_2_/Fe. This fraction also decreased DNA oxidative damage induced by H_2_O_2_ in PBMCs. Regarding the cytotoxicity of saponin fraction (0.5–50 µg/mL) none was found to cause lysis of blood platelets, and PBMCs. Our results, for the first time indicate that saponin fraction from sea buckthorn leaves may be a new promising source of compounds for prophylaxis and treatment of diseases associated with oxidative stress.

## 1. Introduction

Sea buckthorn (*Elaeagnus rhamnoides* (L.) A. Nelson) is a shrub, and it belongs to the *Elaeagnaceae* family [1]. Numerous number of research shows that sea buckthorn organs are source of various chemical compounds, including vitamins (E and C), phenolic compounds, tocopherols, carotenoids, amino acids and fatty acids especially “omega” fatty acids [2,3]. It is commonly known that different preparations, including extracts and fractions from sea buckthorn organs has anti-cancer effect, anti-ulcer activities, hepato-protective properties, antimicrobial and antiviral properties [3].

For many years the problem of various diseases, including cardiovascular diseases and cancers, which are associated with oxidative stress has been investigated [4]. This problem was attacked from various angles. Recent research point to a plants as a source of antioxidant, anticancer and antihemostatic compounds. For example, the results published by Skalski et al. [5,6] points that butanolic extracts, phenolic fraction (in which ellagitannins and flavonoids were the dominant compounds) and non-polar fraction (in which non-polar compounds were the dominant) from leaves of sea buckthorn has antioxidant effect and affects the hemostasis system in vitro [5,6]. Cardiovascular diseases are often associated with obesity, which may be reduced by natural anti-obesity agents. For example, Lee et al. [7] and Pichiah et al. [8] observed that teas from sea buckthorn leaves have anti-obesity properties in vivo. Our earlier research showed that sea buckthorn leaves contained, in addition to phenolics and triterpenoids, also triterpenoid saponins [5]. The purpose of the current study was to determine especially the biological potential of the saponin fraction from sea buckthorn leaves in in vitro different models (using human plasma, human blood platelets and human peripheral blood mononuclear cells). Another goal was to characterize the composition of the fraction. Different biomarkers of oxidation (in human plasma) were measured: (1) lipid peroxidation (determined by thiobarbituric acid reactive substances–TBARS), (2) protein carbonylation, and (3) thiol group level. We have also assessed the effect of this fraction on DNA damage (using the comet assay) in human peripheral blood mononuclear cells (PBMCs). In addition, another aim of our experiments in vitro was to determine the effect of the saponin fraction from sea buckthorn leaves on selected hemostatic parameters of human plasma: the activated partial thromboplastin time (APTT), prothrombin time (PT), and thrombin time (TT). Tested fraction was also evaluated for toxicity against blood platelets (by measuring the extracellular lactate dehydrogenase (LDH) activity)–a marker of platelet damage and against PBMCs using the resazurin reduction assay.

## 2. Material and Methods

### 2.1. Reagents

Dimethyl sulfoxide (DMSO), hydrogen peroxide (H_2_O_2_), 5,5′-dithio-bis(2-nitro-benzoic acid) (DTNB), 2-4-dinitrophenolhydrazine (DNPH), thiobarbituric acid (TBA), versenic acid (EDTA) and Iron (II) sulfate were purchased from Sigma-Aldrich (St. Louis, MO., USA). Trichloroacetic acid (TCA), hydrochloric acid (HCl), sodium chloride (NaCl), ethanol 96%, ethyl acetate, guanidine hydrochloride were acquired from POCh (Gliwice, Poland) and Alfachem (Poznań, Poland). Thrombin was purchased from Biomed (Lublin, Poland). Reagents for PT and APTT were acquired from Kselmed (Grudziąc, Poland).

### 2.2. Plant Material

Sea buckthorn leaves were obtained from a horticultural farm in Sokółka, Podlaskie Voivodeship, Poland (53°24′ N, 23°30′ E). The whole branches of the tested plant were transported to Institute of Soil Science and Plant Cultivation–State Research Institute, Pulawy, Poland (IUNG/HRH/2015/2). The leaves were handpicked, freeze-dried and powdered in a laboratory mill (Retsch ZM200, Haan, Germany).

### 2.3. Preparation of the Saponin Fraction of Sea Buckthorn Leaves

The extract of sea buckthorn leaves was prepared as described previously [5]. Briefly, 248 g of milled sea buckthorn leaves was extracted with 5 L of 80% methanol. The extract was concentrated by rotary-evaporation and defatted by liquid-liquid extraction. After rotary evaporation of organic solvents, the mixture was extracted with n-butanol. The butanol extracts were rotary evaporated to remove the solvent (with addition of MilliQ water), the residue was suspended in portions of water and 20% *tert*-butanol, freezed, lyophilized, and stored in a fridge. A 3.14 g portion of the butanol extract was dissolved in 200 mL of 1.5% methanol + 0.1% formic acid, and sonicated for 5 min. The mixture was centrifuge, and the supernatant was loaded onto a C18 column (65 × 70 mm; Cosmosil 140C18-Prep, 140 µm), equilibrated with 1.5% methanol + 0.1% formic acid. The column was washed with 400 mL of the same solvent, to remove the most polar compounds. Compounds bound to the column were subsequently eluted with methanol solutions of increasing concentration: 10% (300 mL), 30% (500 mL), 66% (500 mL) and 80% (500 mL). The obtained eluates were rotary-evaporated and lyophilized, to yield 0.076 g of 10% methanol fraction (Fr I), 1.427 g of 30% methanol fraction (Fr. II), 0.505 g of 50% methanol fraction (Fr. III), 0.241 g of 66% methanol fraction (Fr. IV) and 0.118 g of 80% methanol fraction (Fr. V). The biological activity and composition of saponin-containing Fr. V has been described in this paper.

### 2.4. LC-MS

The composition of saponin fraction of sea buckthorn leaves was analysed by UHPLC-ESI-MS/MS, using a Thermo Ultimate 3000RS (Thermo FischerScientific, MS, USA) UHPLC system, equipped with a charged aerosol detector (CAD), a diode array detector, and coupled a Bruker Impact II (Bruker Daltonics GmbH, Bremen, Germany) Q-TOF mass spectrometer. Chromatographic separations were performed on a an ACQUITY BEH C18 column (2.1 × 150 mm, 1.7 μm; Waters, MA, USA) maintained at 60 °C, the injection volume was 3.0 μL. Elution was performed with a 26 min concave-shaped gradient (a Thermo gradient curve nr. 6) from 7 to 80% of solvent B (0.1% formic acid in acetonitrile) in solvent A (0.1% formic acid in MilliQ-water), the flow rate was 0.5 mL min^ἂ1^. UHPLC–MS/MS analysis was performed in negative ion mode, the scanning range was from *m/z* 50 to 2000. The following setting were applied: capillary voltage was 3 kV, dry gas flow was 6 L min^−1^, dry gas temperature was 200 °C, nebulizer pressure was 0.7 Bar, collision RF was 700 V, transfer time was 80 ms, prepulse storage time was 10 ms. Two precursor ions of intensity over 2000 counts were fragmented in each scan. Collision energy was set automatically in the range from 3.5 to 140 eV, depending on the *m/z* value of a fragmented ion. The internal calibration of acquired data was performed with sodium formate introduced to the ion source via a 20 µL loop at the beginning of a separation.

### 2.5. Stock Solution

Stock of saponin fraction from sea buckthorn leaves was prepared in 50% DMSO. The final DMSO concentration in the tested samples was less than 0.05%.

### 2.6. Blood Platelets and Plasma Isolation

Whole blood was obtained from healthy volunteers (non-smokers and non-medication). The material was made available by the L. Rydygier Medical Center, in Łódź, Poland. The material was collected in tubes with CPD (citrate/phosphate/dextrose; 9:1; *v/v* blood/CPD). All tests were carried out with the consent of the bioethics commission of the University of Łódź, Poland (resolution number 3/KBBN-U-L/II/2016). None of the subjects had taken any medication or addictive substances such as tobacco, alcohol, antioxidant supplementation or aspirin, or any other anti-platelet drugs. Platelet-rich plasma was prepared by centrifugation of fresh human blood at 1200× *g* for 12 min at room temperature. Blood platelets were then sedimented by centrifugation at 2300× *g* for 15 min at room temperature. The platelet pellet was washed with modified Tyrode’s buffer (pH 7.4). The concentration of the platelets in the suspensions was estimated spectrophotometrically [9] to be 2.0 × 10^8^/mL.

To measure parameters of hemostasis, the plasma or blood platelets were incubated at 37 °C for 30 min with the tested plant fraction (concentration range 0.5–50 µg/mL). To measure the oxidative stress parameters in plasma, the plasma was pre-incubated at 37 °C for 5 min with the tested plant fraction (concentration range 0.5–50 µg/mL) and then treated with 4.7 mM H_2_O_2_/3.8 mM Fe_2_SO_4_/2.5 mM EDTA (25 min, at 37 °C).

### 2.7. PBMCs Isolation

Peripheral blood mononuclear cells were isolated from the leucocyte-buffy coat collected from blood of healthy, non-smoking donors from Blood Bank (Lodz, Poland). The leucocyte-buffy coat was diluted in a 1:1 ratio in 1% phosphate buffer saline (PBS) and centrifuged in a density gradient of Lymphosep (Cytogen, Zgierz, Poland) at 200× *g* for 20 min at room temperature. Then PBMCs cells were collected and washed three times by centrifugation in 1% PBS. The pellet of the cells was resuspended in RPMI 1640 medium (Lonza, Bazel, Switzerland) [10].

### 2.8. Markers of Oxidative Stress

#### 2.8.1. Lipid Peroxidation Measurement

500 µL TCA and TBA were added to the post-incubation samples. The sealed tubes were vortexed. Holes have been made in the eppendorf caps for venting. All test samples were boiled (10 min, 100 °C). After cooling, the samples were centrifuged (10,000 rpm, 15 min, 18 °C). The supernatant absorbance at λ = 535 nm was measured. The TBARS concentration was calculated using the molar extinction coefficient (ε = 156,000 M^−1^cm^−1^) [5,11].

#### 2.8.2. Carbonyl Group Measurement

Test samples were diluted in 450 μL 0.9% NaCl. 500 µL 40% TCA was added to the diluted samples. After the incubation, the samples were centrifuged (2500 rpm, 5 min, 4 °C). The supernatant has been removed. 10 mM DNPH solution was added to the samples. Samples were vortexed and incubated for 1 h in the dark. After completed incubation, 750 µL of 40% TCA was added to the samples to re-protein. The samples were centrifuged (2500 rpm, 5 min, 4 °C) and the supernatant was removed. 1.5 mL of mixed ethanol-ethyl acetate (1:1) was added to the samples, the samples were mixed and centrifuged (2500 rpm, 5 min, 4 °C) and the supernatant was removed (this procedure was repeated three times). Guanidine hydrochloride was added to the precipitate. The precipitate was dissolved on a shaker. The absorbance of the samples was measured at λ = 375 nm. The carbonyl group concentration was calculated using a molar extinction coefficient (ε = 22,000 M^−1^cm^−1^) [5,11].

#### 2.8.3. Thiol Group Determination

The level of thiol groups was measured by spectrophotometry. 20 µL samples were added on 96-well plate, then 20 µL SDS and 160 µl phosphate buffer (pH 8) were added. The absorbance of A_0_ was measured at λ = 412 nm. Then 16.6 µL DTNB was added to the tests samples and after one hour incubation (37 °C) the absorbance A_1_ was measured again. The absorbance difference A_1_−A_0_ was calculated. The thiol group concentration was calculated using a molar extinction coefficient (ε = 13,600 M^−1^cm^−1^) [5,12,13].

#### 2.8.4. DNA Damage by the Comet Assay

The comet assay was performed under alkaline conditions according to the procedure of Tokarz et al. [14]. Saponin fraction was added to the suspension of PBMCs to give final concentrations of 0.5, 1, 5, 10 and 50 μg/mL. PBMCs were incubated for 2 h at 37 °C in 5% CO_2_. After treatment with saponin fraction, PBMCs were washed and suspended in RPMI 1640 medium. Then H_2_O_2_ was added at the concentration of 25 μM. PBMCs were incubated for 15 min on ice. After incubation, a freshly prepared suspension of the cells in 0.75% LMP agarose dissolved in PBS was spread onto microscope slides which were pre-coated with 0.5% NMP agarose. Then, the cells were lysed for 1 h at 4 °C in a buffer containing 2.5 M NaCl, 0.1 M EDTA, 10 mM Tris, 1% Triton X-100, pH 10. After cells lysis, the slides were placed in an electrophoresis unit. DNA was allowed to unwind for 20 min in the solution containing 300 mM NaOH and 1 mM EDTA, pH > 13.

Electrophoretic separation was performed in the solution containing 30 mM NaOH and 1 mM EDTA, pH > 13 at ambient temperature of 4 °C (the temperature of the running buffer did not exceed 12 °C) for 20 min at an electric field strength of 0.73 V/cm (28 mA). Then, the slides were washed in water, drained, stained with 2 µg/mL DAPI and covered with cover slips. In order to prevent additional DNA damage, the procedure described above was conducted under limited light or in the dark.

#### 2.8.5. Comets Analysis

The comets were observed at 200× magnification in an Eclipse fluorescence microscope (Nikon, Japan) attached to a COHU 4910 video camera (Cohu, Inc., San Diego, CA, USA) equipped with a UV-1 A filter block and connected to a personal computer-based image analysis system Lucia-Comet v. 7.3 (Laboratory Imaging, Praha, Czech Republic). Fifty images (comets) were randomly selected from each sample and the mean value of DNA in comet tail was taken as an index of DNA damage (expressed in percent).

### 2.9. Parameters of Hemostasis

#### 2.9.1. The measurement of PT

Prothrombin time (PT) was determined using a coagulometer (K-3002 OPTIC), according to the method described by Malinowska et al. [15].

#### 2.9.2. The Measurement of APTT

Activated partial thromboplastin time (APTT) was determined using a coagulometer (K-3002 OPTIC), according to the method described by Malinowska et al. [15].

#### 2.9.3. The Measurement of TT

Thrombin time (TT) was determined using a coagulomenter (K-3002 OPTIC), according to the method described by Malinowska et al. [15].

### 2.10. Parameters of Toxicity

#### 2.10.1. LDH Measurement

270 µL of phosphate buffer (pH 7.4) was added to a 96 well plate, then 10 µL of previously prepared test samples and 10 µL of NADH solution were added. The 96-well plate was incubated at room temperature for 20 min. 10 μL of pyruvate solution was added to the wells of the plate and the measurement of absorption at λ = 340 nm was immediately started. LDH activity was expressed as U [16].

#### 2.10.2. Cell Viability

The cell viability resazurin assay was performed similar to described by O’Brien et al. [17] Resazurin salt powder was dissolved in sterile PBS buffer. Cells were seeded on the 96-well plates in count of 50 000 per well. Saponin fraction was added to wells to obtain a final concentrations of 0.5, 1, 5, 10 and 50 µg/mL and incubated for 2 h, 6 h and 24 h at 37 °C in 5% CO_2_. Following 10 µL of resazurin salt was added to each well and plates again were incubated at 37 °C in 5% CO_2_ for 2 h. Next fluorescence was measured with HT microplate reader Synergy HT (BioTek Instruments, USA) using an λ_ex_ = 530/25 and an λ_em_ = 590/35 nm. Saponin fraction effects were quantified as the percentage of control fluorescence.

## 3. Results

Results of LC-MS/MS analysis of the saponin fraction of sea buckthorn leaves are shown in Table 1; several small peaks were not listed in the table, when no fragmentation data was obtained. As it might be expected, putative triterpenoid saponins were dominant constituents of the saponin fraction of sea buckthorn leaves, and contributed to about 68.3% of the total CAD peak area (Table 1). It is possible to observe chains of neutral losses of monosaccharides, as well as ions of potential triterpenoid aglycones in CID spectra of 18 constituents of the saponin fraction of sea buckthorn leaves, which confirms the tentative identification of these compounds. Ten of these compounds was additionally acylated with acetic acid and/or a monoterpenoids acid (probably (-)-linalool-1-oic acid), as indicated by the observed neutral losses of 42, 60 and 166 Da fragments, corresponding (as confirmed by HRMS) to CH_2_CO, CH_3_COOH, and C_10_H_14_O_2_, respectively. In addition, an ion at *m/z* 183, corresponding to C_10_H_15_O_3_, was always observed in fragmentation spectra of the acylated saponins.

A few free terpenoids were also found (20.9% of the total peak area), but among them a single C_30_H_48_O_5_ compound vastly dominated. Acylated triterpenoids were sparse, and present at low levels (about 1.0% of the total peak area). The fraction also contained small amounts of lysophospholipids: lysophosphatidylinositols, lysophosphatidylglycerols, and lysophosphatidic acids; they contributed to about 2.5% of the total peak area. More hydrophilic compounds were represented mainly by ellagic acid, its pentoside, and acylated isorhamnetin glycosides (about 1.2% of the total peak area).

Human plasma treated with saponin fraction from sea buckthorn leaves (50 µg/mL) demonstrated statistically significant reduction of plasma lipid peroxidation induced by H_2_O_2_/Fe compared to positive control (plasma treated with only H_2_O_2_/Fe), with 25% inhibition (*p* < 0.05). However, no change in the level of plasma lipid peroxidation induced by H_2_O_2_/Fe was observed for other used concentrations of saponin fraction (0.5–5 µg/mL) (Figure 1A). Plasma protein carbonylation induced by H_2_O_2_/Fe was also inhibited by the highest tested concentrations of saponin fraction (10 and 50 µg/mL) (*p* < 0.05) (Figure 1B). The greatest inhibition was demonstrated by saponin fraction at the highest tested concentration (50 µg/mL), with decrease of about 30% compared to positive control (plasma treated with only H_2_O_2_/Fe) (Figure 1B). On the other hand, no change in thiol group level was observed in any of the treated plasma stimulated by H_2_O_2_/Fe in the presence of saponin fraction (Figure 1C). Similar results were observed for coagulation times (data not shown).

Figure 2 demonstrates the level of DNA damage analyzed by the comet assay under alkaline conditions. The comet assay in the alkaline version is the sensitive and simple method determining of DNA damage level including single- and double-strand breaks and alkali-labile sites in living cells [18]. Saponin fraction from sea buckthorn leaves at the concentrations of 1, 5 and 10 µg/mL induced small DNA damage up to 10% (Figure 2). Interestingly, DNA damage did not increase with increasing concentration of the saponin fraction. It was also shown that saponin fraction at the concentrations of 1, 5, 10 and 50 µg/mL decreased DNA oxidative damage induced by H_2_O_2_ in PBMCs (Figure 2). Figure 3 shows representative pictures of the comets obtained after pre-incubation of PBMCs with saponin fraction at different concentrations and followed incubation with H_2_O_2_.

Regarding the cytotoxicity of saponin fraction (0.5–50 µg/mL) none was found to cause lysis of blood platelets (Figure 4A). Similarly, for PBMCs no cytotoxic effect was observed after incubation for 2, 6 and 24 h with saponin fraction (Figure 4B–D).

## 4. Discussion

In contrast to carotenoids, phytosterols, tocopherols and tocotrienols, other hydrophobic specific metabolites of sea buckthorn have been poorly characterized. It seems that until recent years, literature data about the presence of saponins in leaves of sea buckthorn were hardly available, and was based only on simple laboratory tests [19]. Our previous publications [5,20], as well as the present work provide some more detailed information about these compounds. Eighteen putative saponins were detected, including 10 compounds acylated with a C_10_H_16_O_3_ monoterpenoid acid. The saponins detected in the fraction, in any case the acylated ones, are probably new compounds. Sea buckthorn were shown to contain Yang et al. [21] isolated glycosides of isorhamnetin and kaempferol acylated with (-)-linalool-1-oic acid from the leaves of sea buckthorn, so it seems to be very probable that the detected saponins are acylated with the same acid. Zheng et al. [22] analyzed extracts of air-dried fruit, leaves and twigs of sea buckthorn by LC-HRMS/MS, and detected the presence of compounds giving deprotonated ions at *m/z* 1459, and 1313 in the negative ion mode. They did not present MS/MS fragments for these substances, and were not able to identify them. The highest levels of these compounds were observed in the fruit. In this case, Zheng et al. [22] most probably detected saponins from sea buckthorn seeds. Our preliminary analyses showed that sea buckthorn seeds contained the same saponins as those from the leaves (data not shown). Moreover, triterpenoid saponins of different kind were earlier isolated from seeds of *Hippophae rhamnoides* ssp. *sinensis* [23,24]. The saponin fraction contained also significant amounts of triterpenoids. While presence in of such compounds as pomolic acid, and dulcioic acid (C_30_H_48_O_4_), acylated triterpenoids, and especially oleanolic acid, and ursolic acid (C_30_H_48_O_3_) in sea buckthorn was shown by different researchers [25,26,27], C_30_H_48_O_5_ triterpenoids were most probably reported only in our earlier publications on sea buckthorn, and in the present work [5,28]. Literature data have demonstrated that steroidal and triterpenoid saponins isolated from different plants have anti-platelet and anticoagulant potential in in vitro and in vivo models [29,30,31,32]. For example, Xiong et al. [32] have observed anticoagulant properties of saponins form *Panax notoginseng* root and rhizome (at concentrations: 0.5–3.5 mg/mL, in vitro). However, we did not observed anticoagulant properties of saponin fraction (using lower concentrations: 0.5–50 µm/mL) from sea buckthorn leaves. 

Oxidative stress is considered as one of the key mechanisms associated with human pathological processes, including cardiovascular diseases and cancers. Zhang et al. [33] have described that saponins isolated from the roots and rhizomes of *Panax notoginseng* may play an important role in treating different cardiovascular diseases, including angina pectoris, coronary heart diseases, and arrhythmia cordis. Authors suggest that the main bioactive ingredients are dammarane triterpene saponins. Cardioprotective action (especially anti-atherosclerosis properties) of 20(S)-protopanaxadiol saponins was also observed in ApoE deficient mice [31] In our present experiments, saponin fraction isolated from sea buckthorn leaves (especially at the highest used concentrations: 10 and 50 µg/mL) showed anti-oxidative against harmful to human plasma protein and lipid oxidation induced by H_2_O_2_/Fe. Dong et al. [34] observed also that *P. notoginseng* saponins has protected the human umbilical vein endothelial cells from H_2_O_2_—induced oxidative stress. Results of Xing et al. [35] demonstrated that saponins (125 and 250 mg/kg daily, for 15 days) from leaves of *P. quinquefolius* exert significant effects on cisplatin–induced cardiotoxicity in part by inhibition of nuclear factor kappa-light-chain-enhancer of activated B cells (NF-ĸB) activity and regulation of phosphoinositide 3-kinase (PI3K)/protein kinase B (Akt)/apoptosis mediated signaling pathways, in mice. In addition, saponins reduced the oxidative stress induced by cisplatin. Authors measured the level of oxidative stress by various parameters, including the level of reactive oxygen species, the level of glutathione and superoxide dismutase activity. The similar action was observed by Wu et al. [36]. In this experiment, authors used steroidal saponins extract from *Ophiopogan japonicas* root. This extract ameliorated doxorubicin-induced chronic heart failure by inhibiting oxidative stress in rat model. Hu et al. [37] also observed antioxidantive potential of *P. notogineng* saponins. 

It was shown that garlic saponins prevented DNA damage induced by H_2_O_2_ in mouse-derived C2C12 myoblasts [38]. Moreover, garlic saponins prevented H_2_O_2_-induced growth inhibition of these cells and decreased the level of intracellular ROS. The data obtained by Kang et al. indicate that garlic saponins activate the Nrf/HO-1 pathway and by this way induce of antioxidative and detoxifying enzymes in C2C12 myoblasts. Similarly, it was found the protective effects of Timosaponin AIII (TA-III), a naturally occurring steroidal saponin from *Anemarrhena asphodeloides,* against UVB-mediated formation of 8-oxo-7,8-dihydro-2′-deoxyguanosine (8-oxo-dG) in human epidermal keratinocytes [39]. It was also revealed that TA-III activated DNA repair enzymes and cell cycle arrest genes like PCNA (proliferating cell nuclear antigen) and SMC1 (structural maintenance of chromosomes protein 1). 

On the other hand it was found that saponins can be cyto- and genotoxic for many cancer cells. Crude saponins from fruit, bark and leaves of *Zanthoxylum armatum* DC. (Zf.Sa, Zb.Sa and ZI.Sa) showed significant growth inhibition against human breast (MCF-7, MDA-MB-468) and colorectal (Caco-2) cancer cell lines [40]. Saponins from fruits induced apoptosis in all three types of cell lines while saponins from leaves and bark showed similar results against MDA MB-468 cells, it was seen by nuclear fragmentation and chromatin condensation with DAPI staining. Saponin 6 derived from *Anemone taipaiensis* induced cell apoptosis and cell cycle arrest in U87 human malignant glioblastoma cells [41]. 

This study provides for the first time information about chemical content and biological properties of saponin fraction from sea buckthorn leaves, indicating that this fraction could be used as valuable and safe source of antioxidants. We suppose that especially triterpenoid saponins may be compounds with antioxidant properties.

## Figures and Tables

**Figure 1 molecules-25-03004-f001:**
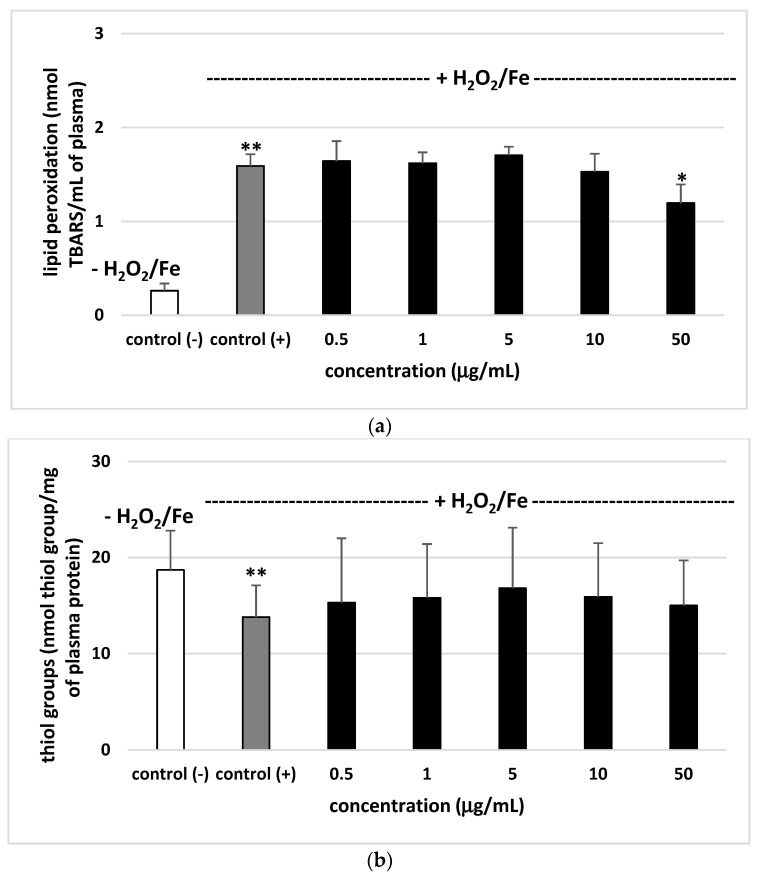
The effect of saponin fraction from the leaves of *Elaegnus rhamnoides* (L.) A. Nelson (0.5–50 µg/mL) on plasma lipid peroxidation induced by H_2_O_2_/Fe (**A**), on the oxidation of protein thiols induced by H_2_O_2_/Fe (**B**) and on carbonyl group formation (plasma protein oxidation) induced by H_2_O_2_/Fe (**C**). Data represents means ± SD of 6 experiments (from different donors). ** *p* < 0.01 (for control (+) vs. control (−)), * *p* < 0.05 vs. control (+H_2_O_2_/Fe). Test: ANOVA.

**Figure 2 molecules-25-03004-f002:**
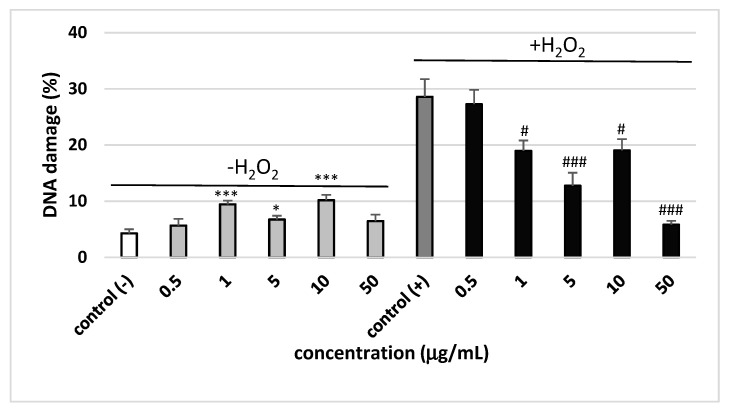
The effect of saponin fraction from the leaves of *Elaegnus rhamnoides* (L.) A. Nelson (0.5–50 µg/mL) on DNA damage in PBMCs treated with H_2_O_2_ at 25 µM for 15 min on ice. Data represents means ± SEM of 6 experiments (from different donors). * *p* < 0.05, ** *p* < 0.01, *** *p* < 0.001 (for control (−) vs. saponin fraction samples/−H_2_O_2_); ^#^
*p* < 0.05, ^###^
*p* < 0.001 (for +H_2_O_2_ vs. saponin fraction samples/+H_2_O_2_). Test: ANOVA.

**Figure 3 molecules-25-03004-f003:**
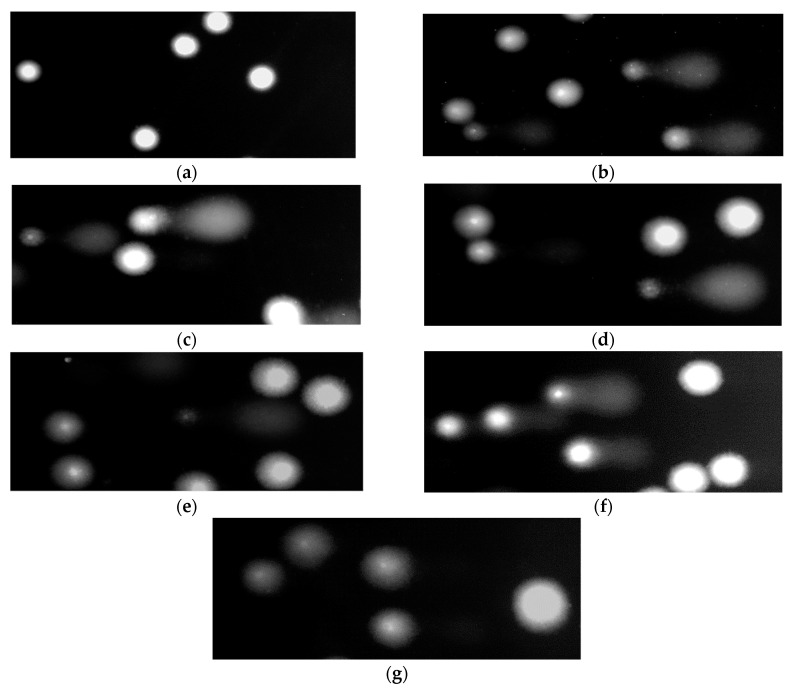
Representative photos of comets obtained in the alkaline version of the comet assay after pre-incubation of PBMCs with saponin fraction from the leaves of *Elaegnus rhamnoides* (L.) A. Nelson (0.5–50 µg/mL) and incubation with 25 µM H_2_O_2_ for 15 min on ice. (**A**)–control (untreated PBMCs); (**B**)–PBMCs incubated with H_2_O_2_ at 25 µM for 15 min on ice; (**C**–**G**)–PBMCs pre-incubated with saponin fraction at the concentrations from 0.5 to 50 µg/mL, respectively and then incubated with H_2_O_2_ at 25 μM for 15 min on ice.

**Figure 4 molecules-25-03004-f004:**
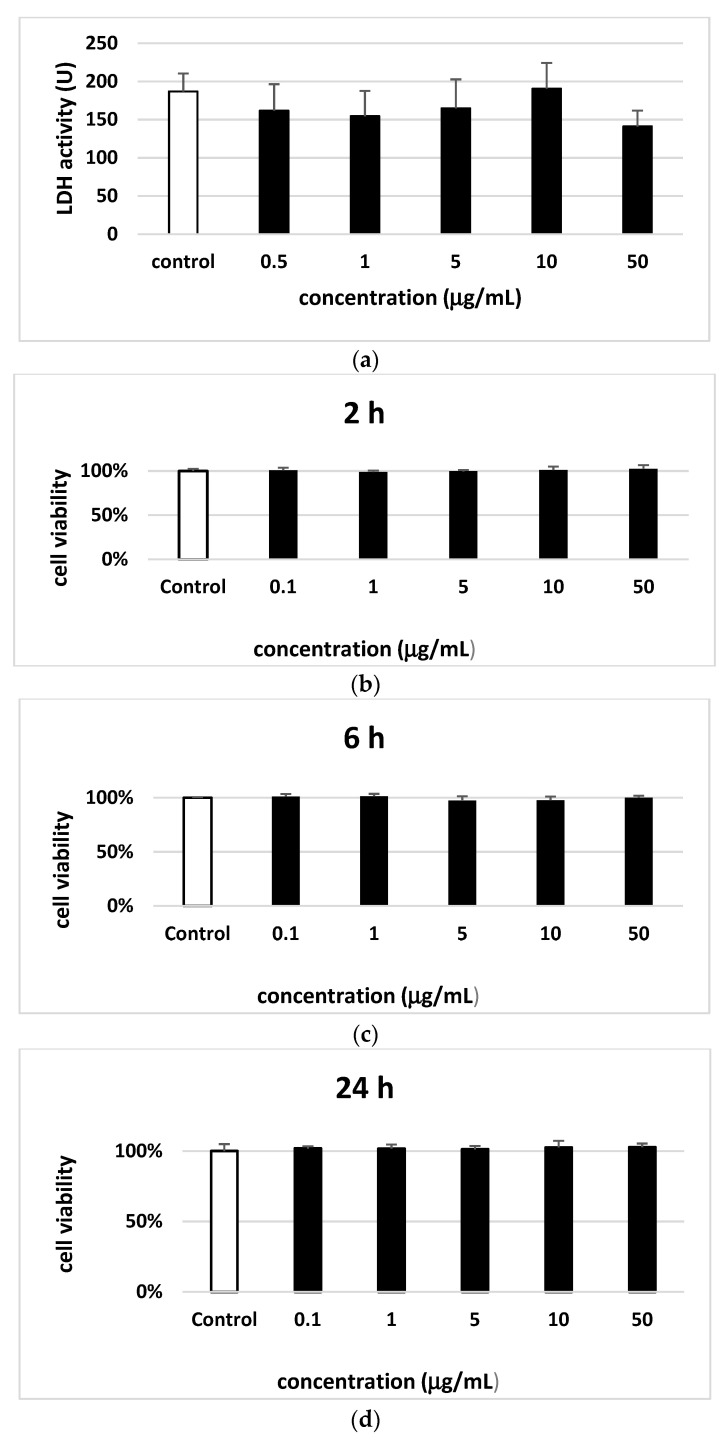
The effect of saponin fraction from the leaves of *Elaegnus rhamnoides* (L.) A. Nelson (0.5–50 µg/mL) on the damage of human blood platelets (**A**) and on the viability of PBMCs. The cell viability for blood platelets was measured after 30 min. The cell viability for PBMCs was measured after 2 h (**B**), 6 h (**C**) and 24 h (**D**) incubation with saponin fraction. Data represents means ± SD of 6 experiments (from different donors). *P* < 0.05 vs. control. Test: ANOVA.

**Table 1 molecules-25-03004-t001:** Specific metabolites in the saponin fraction of sea buckthorn leaves.

	tR	Peak Area Fraction (%)	Parent Ion [M-H]^−^	MM/MS Fragments	Formula	Error (ppm)	mΣ	Tentative Identification
1	0.84	0.37	317.0547	225.0081 (8)				unidentified
			245.0430	171.0050 (9), 152.9962 (10)	C_6_H_15_O_8_P	0.7	3.5	glycerophosphoglycerol
2	5.02	0.13	433.0420	300.9992 (100)	C_19_H_14_O_12_	−1.8	13.1	ellagic acid-Pen
3	5.38	0.7	300.9998		C_14_H_6_O_8_	−2.8	4.3	ellagic acid
4	13.21	0.13	789.2613	626.2026 (11), 477.1050 (42), 313.0371 (100), 300.0283 (63)	C_38_H_46_O_18_	−0.2	12.1	I-dHex-Lin-Hex
			489.2714	327.2213 (15), 309.2073 (100), 171.1044 (21)				unidentified
5	13.33	0.20	789.2613	626.2017 (10), 477.1049 (22), 313.0362 (100), 300.0280 (33)	C_38_H_46_O_18_	−0.2	8.8	I-dHex-Lin-Hex
6	13.67	0.47	1119.5593 *	1073.5551 (34),911.5017 (5), 749.4499 (9), 603.3913 (30), 585.3799 (24), 471.3485 (100)	C_53_H_86_O_22_	−0.0	20.8	C_30_H_48_O_4_-Pen-dHex-Hex-Hex
7	13.79	0.14	927.4971	765.4453 (19), 603.3931 (68), 585.3795 (16), 471.3465 (36)	C_47_H_76_O_18_	−1.3	26.4	C_30_H_48_O_4_-Pen-Hex-Hex
8	14.11	0.29	511.2406 *	465.2338 (13), 293.0889 (100), 171.1405 (49)	C_21_H_38_O_11_	−1.9	3.7	
			927.4959	765.4440 (22), 603.3922 (100), 585.3794 (23), 471.3491 (55)	C_47_H_76_O_18_	−0.0	25.0	C_30_H_48_O_4_-Pen-Hex-Hex
9	14.39	1.51	695.4018 *	649.3976 (2), 487.3441 (100)	C_36_H_58_O_10_	−0.8	10.7	C_30_H_48_O_5_-Hex
10	14.51	0.67	1427.6703 *	1219.6117 (29), 1057.5600 (2), 733.4544 (20), 587.3969 (27), 569.3862 (24), 455.3540 (100)	C_65_H_106_O_31_	−0.2	15.5	C_30_H_48_O_3_-Pen-dHex-Hex-Hex-Hex-Hex
11	14.95	4.80	713.3323 **	733.4534 (85), 587.3959 (74), 569.3854 (34), 455.3544 (100)	C_65_H_106_O_31_	−1.4	8.8	C_30_H_48_O_3_-Pen-dHex-Hex-Hex-Hex-Hex
12	15.38	1.01	1235.6069	749.4480 (4), 587.3962 (100), 569.3869 (13), 455.3533 (20)	C_59_H_96_O_27_	−0.3	35.5	C_30_H_48_O_3_-Pen-Hex-Hex-Hex-Hex
13	15.59	0.30	503.3385		C_30_H_48_O_6_	−1.3	3.3	triterpenoid
14	15.78	0.16	489.2140	327.1618 (100), 312.1378 (78), 297.1141 (45), 255.0678 (32)	C_26_H_34_O_9_	−2.1	10.5	unidentified
15	16.14	0.15	491.2301	329.1773 (100), 314.1542 (38)	C_26_H_36_O_9_ ?	−2.9	15.4	unidentified
16	16.64	4.99	1265.6167 *	895.5050 (3), 733.4528 (43), 587.3972 (39), 569.3850 (36), 455.3538 (100)	C_59_H_96_O_26_	0.4	17.0	C_30_H_48_O_3_-Pen-dHex-Hex-Hex-Hex
17	16.86	8.97	1459.7114	1297.6621 (8), 973.5561 (15), 807.4554 (14), 661.3975 (22), 619.3858 (8), 601.3741 (25), 529.3548 (70), 487.3443 (12), 469.3341 (60), 183.1039 (32)	C_71_H_112_O_31_	0.0	17.6	C_30_H_48_O_5_-Ac-Pen-dHex-Lin-Hex-Hex-Hex
18	17.12	3.13	1313.6537	1151.5990 (5), 827.4988 (6), 809.4857 (6), 661.3968 (9), 619.3840 (5), 601.3760 (36), 487.3448 (2), 469.3329 (10), 183.1039 (14)	C_65_H_102_O_27_	−0.1	15.8	C_30_H_48_O_5_-Ac-Pen-Lin-Hex-Hex-Hex
19	17.32	6.89	1313.6528	985.4993 (8), 823.4497 (11), 661.3964 (100), 619.3864 (9), 601.3751 (47), 529.3559 (7), 487.3426 (3), 469.3329 (10), 183.1031 (26)	C_65_H_102_O_27_	0.6	30.9	C_30_H_48_O_5_-Ac-Pen-Hex-Lin-Hex-Hex
20	17.91	0.26	329.1399	311.1294 (72), 275.0936 (1), 163.0768 (100), 149.0609 (87)	C_19_H_22_O_5_	−1.3	3.5	unidentified
21	18.13	11.34	1401.7034	1239.6516 (4), 749.4478 (27), 603.3893 (39), 585.3804 (30), 471.3486 (100), 183.1032 (12)	C_69_H_110_O_29_	1.9	18.8	C_30_H_48_O_4_-Pen-dHex-Lin-Hex-Hex-Hex
22	18.26	7.07	1343.6633 *	1297.6574 (100), 1135.6026 (17), 973.5552 (11), 807.4559 (10), 661.3972 (17), 619.3882 (10), 601.3747 (18), 529.3530 (57), 487.3442 (5), 469.3333 (41), 183.1027 (27)	C_65_H_102_O_26_	0.6	39.4	C_30_H_48_O_5_-Ac-Pen-dHex-Lin-Hex-Hex
			551.3442 *	505.3386 (100), 343.2872 (19), 179.0563 (18)	C_26_H_50_O_9_	−0.9	8.6	unidentified glycoside
23	18.46	1.82	1151.5998	989.5454 (12), 985.5001 (16), 823.4575 (6), 661.3974 (7), 619.3875 (5), 601.3728 (25), 487.3423 (2), 469.3318 (19), 183.1045 (9)	C_59_H_92_O_22_	0.8	16.7	C_30_H_48_O_5_-Ac-Pen-Hex-Lin-Hex
			315.1606		C_19_H_24_O_4_	−1.0	11.2	unidentified
			551.3433 *	505.3389 (100), 343.2860 (9), 291.1442 (5)				unidentified
24	18.67	2.30	1151.5995	989.5474 (14), 985.5027 (15), 823.4501 (7), 661.3946 (13), 619.3864 (4), 601.3744 (9), 501.3213 (17), 487.3432 (1), 469.3317 (7), 183.1030 (11)	C_59_H_92_O_22_	1.1	19.3	C_30_H_48_O_5_-Ac-Pen-Hex-Lin-Hex
25	18.80	7.40	1255.6457	1093.5980 (7), 769.4888 (5), 603.3899 (100), 585.3797 (23), 471.3473 (27), 183.1032 (12)	C_63_H_100_O_25_	1.9	17.2	C_30_H_48_O_4_-Pen-Lin-Hex-Hex-Hex
26	19.24	20.33	487.3431	409.3115 (5)	C_30_H_48_O_5_	−0.4	2.7	triterpenoid
27	19.97	4.11	1239.6514	1077.5965 (8),915.5455 (4), 749.4484 (14), 603.3897 (37), 585.3785 (30), 471.3486 (100), 183.1027 (12)	C_63_H_100_O_24_	1.4	36.0	C_30_H_48_O_4_-Pen-dHex-Lin-Hex-Hex
28	20.20	0.15	649.3741	163.0394 (5), 145.0293 (31), 117.0340 (10)	C_39_H_54_O_8_	0.7	8.6	C_30_H_48_O_6_-CouA
29	20.32	0.30	487.3429		C_30_H_48_O_5_	0.0	2.9	triterpenoid
30	20.53	0.80	1093.5944	931.5389 (13), 765.4373 (6), 603.3888 (100), 585.3797 (25), 471.3474 (45), 183.1034 (10)	C_57_H_90_O_20_	0.8	15.7	C_30_H_48_O_4_-Pen-Hex-Lin-Hex
31	20.67	0.39	1057.5570	895.5035 (6), 733.4526 (7), 587.3952 (24), 569.3857 (14), 455.3534 (100)	C_53_H_86_O_21_	1.8	13.6	C_30_H_48_O_3_-Pen-dHex-Hex-Hex
32	21.19	2.98	329.1760	314.1528 (45)	C_20_H_26_O_4_ ?	−0.5	3.8	unidentified
			473.3641	413.3427 (17)	C_30_H_50_O_4_	−1.0	3.1	triterpenoid
33	21.98	0.32	293.2125	275.2018 (100), 171.1034 (17)	C_18_H_30_O_3_	−1.1	5.0	unidentified
34	22.61	0.63	633.3798	455.3163 (8), 163.0404 (1), 145.0300 (26), 117.0352 (6)	C_39_H_54_O_7_	−0.1	3.2	C_30_H_48_O_5_-CouA
			571.2894	315.0504 (25), 255.2336 (100), 241.0129 (29), 223.0020 (9), 152.9971 (10)	C_25_H_49_O_12_P	−0.8	10.1	palmitoyl-glycerophosphoinositol
35	22.95	0.21	633.3794	145.0295 (27), 117.0343 (8)	C_39_H_54_O_7_	0.4	7.6	C_30_H_48_O_5_-CouA
36	23.21	0.39	571.2889	391.2265 (5), 315.0491 (8), 255.2332 (53), 241.0122 (18), 152.9957 (5)	C_25_H_49_O_12_P	−0.1	7.5	palmitoyl-glycerophosphoinositol
			481.2575	253.2177 (82), 245.0438 (100), 171.0073 (8)	C_22_H_43_O_9_P	−0.5	9.9	hexadecenoyl-glycerophosphoglycerol
37	23.42	0.20	507.2734	279.2336 (100), 152.9956	C_24_H_45_O_9_P	−1.0	11.1	octadecadienoyl-glycerophosphoglycerol
			471.3483		C_30_H_48_O_4_	−0.7	8.0	triterpenoid
38	23.72	1.19	555.2854	225.0077				unidentified
			481.2577	253.2178 (22), 152.9958 (7)	C_22_H_43_O_9_P	−1.1	11.4	hexadecenoyl-glycerophosphoglycerol
39	24.33	0.55	433.2363	279.2326 (11), 152.9963 (28)	C_21_H_39_O_7_P	−0.4	10.3	octadecadienoyl-glycerolphosphate
40	25.25	0.19	409.2358	25.2339 (12), 152.9956 (30)	C_19_H_39_O_7_P	0.6	10.7	palmitoyl-glycerolphosphate

* M+FA-H^−^; ** M+FA-2H^−^; I–isorhamnetin; dHex–deoxyhexose; Hex–hexose; Ac–acetic acid; Lin–(−)-linalool-1-oic acid; Pen–pentose; Cou–*p*-coumaric acid; mΣ–isotopic pattern fit factor, the lower, the better fit.

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
