# Peer review of "LC/MS Analysis of Saponin Fraction from the Leaves of Elaeagnus rhamnoides (L.) A. Nelson and Its Biological Properties in Different In Vitro Models"

_molecules, 2020, doi:10.3390/molecules25133004_

Round 1
Reviewer 1 Report
This manuscript descripted the LC/MS analysis and the biological properties of leaves of Elaeagnus rhamnoides. It may be important for the plant. My suggestion are listed below.
- The title is "Chemical analysis of saponin fraction...", but actualy only LC/MS data was used. So the title may be changed to "LC/MS anslysis of of saponin fraction...".
- Table 1, the width of some column are too small, such as No, tR, parent ion, and formula.
Author Response
This manuscript descripted the LC/MS analysis and the biological properties of leaves of Elaeagnus rhamnoides. It may be important for the plant. My suggestion are listed below.
Response:
We thank the reviewer for helpful comments. Moreover, authors agree with the comment of Reviewer, and this wrong statement was corrected.
- The title is "Chemical analysis of saponin fraction...", but actualy only LC/MS data was used. So the title may be changed to "LC/MS anslysis of of saponin fraction...".
Response: We have corrected. Now, it is “LC/MS analysis of saponin fraction from the leaves of Elaeagnus rhamnoides (L.) A. Nelson and its biological properties in different in vitro models”
- Table 1, the width of some column are too small, such as No, tR, parent ion, and formula.
Response: Columns of the revised table should have proper widths.

Reviewer 2 Report
This study is concerned with the analysis of the sapomin fraction of the leaves from the Sea buckthorn plant. This work involved analysis of the chemical composition of the saponin fraction using LC-MS, as well as the biological testing of this fraction in a number of assays.
The introduction is informative, but is quite brief and additional information and detail would be preferable. The materials and methods appear to be thorough and well-described.
For the analysis of the chemical composition, a number of the peaks from the LC-MS analysis were able to be identified, although I do note that the authors, while they identify compounds by their class, i.e. as "triterpenoid saponin" etc, they do not specify the identity of the actual compound. It would be ideal if this were able to be done or at least a comment about this made in the main manuscript.
The authors should include a Figure with the chemical structures of the compounds that they do specifically identify as being part of the saponin fraction in the Table.
For the biological analysis, when the authors discuss statistically-significant differences (i.e. lipid peroxidation), they should also quote the p-values in the text.
For the biological analysis, it appears that the authors analysed the activity of the entire fraction. It would be of particular interest to identify what specific compounds in the the saponin fraction were actually responsible for the activity that has been identified. Furthermore, in the discussion, there is supposition that the activity seen by the fraction is due to triterpenoid saponins, but this is based off of previous literature and not the research presented in this paper.
In the DNA damage assay, it can be seen that DNA damage did not necessarily increase with increasing concentration of the saponin fraction - this is unusual and should be commented on by the authors.
Overall, there is merit in this work, although I find that the conclusions that can be made are somewhat limited - while the authors have endeavoured to provide information about the chemical composition of this saponin fraction and also identify biological activity of this fraction, they do not appear to go that step further to attempt to identify what specific components are responsible for the activity seen - this limits the impact of this work.
Author Response
This study is concerned with the analysis of the sapomin fraction of the leaves from the Sea buckthorn plant. This work involved analysis of the chemical composition of the saponin fraction using LC-MS, as well as the biological testing of this fraction in a number of assays.
Response:
We thank the reviewer for helpful comments. Moreover, authors agree with the comment of Reviewer, and this wrong statement was corrected.
The introduction is informative, but is quite brief and additional information and detail would be preferable. The materials and methods appear to be thorough and well-described.
Response: Information about the presence of triterpenoid saponins in sea buckthorn leaves has been added to the Introduction.
For the analysis of the chemical composition, a number of the peaks from the LC-MS analysis were able to be identified, although I do note that the authors, while they identify compounds by their class, i.e. as "triterpenoid saponin" etc, they do not specify the identity of the actual compound. It would be ideal if this were able to be done or at least a comment about this made in the main manuscript.
Response: We had decided to label the detected saponins as merely “triterpenoid saponins” because it was difficult to distinguish the actual aglycone ions for some of the acylated compounds. It should be underlined that the detected sea buckthorn saponins are most probably new compounds, with no literature data available. In addition, we feared that long “names” for tentatively identified compounds would unnecessarily enlarge Table 1. As a result, we had decided to apply an uniform and relatively brief labelling “triterpenoid saponins” to all saponins. The revised version of Table 1 provides all details of structures of the saponins which could be deduced from HRMS/MS spectra. Additional comments about structures of the saponins have been added to the Results and the Discussion.
The authors should include a Figure with the chemical structures of the compounds that they do specifically identify as being part of the saponin fraction in the Table.
Response: Unfortunately, we lack necessary data about actual structure of constituents of the saponin fraction, they were only tentatively identified on the basis of their HRMS/MS spectra. Most of the detected sea buckthorn putative saponins, possibly all of them, are new compounds. None of these compounds has been purified so far, we do have no precise information about their structures, even about their aglycones (except their molecular formulas). We could present the structure of ellagic acid, but it was only a minor constituent of the fraction (below 1% of the total CAD peak area); in contrast, it is not possible to show the structure of the detected ellagic acid pentoside, as the place of t pentose substitution (and a type of pentose) is unknown. A figure of ellagic acid could unnecessarily suggest that it is a one of major constituents of the fraction.
For the biological analysis, when the authors discuss statistically-significant differences (i.e. lipid peroxidation), they should also quote the p-values in the text.
Response: We have added the p-values in the text
For the biological analysis, it appears that the authors analysed the activity of the entire fraction. It would be of particular interest to identify what specific compounds in the the saponin fraction were actually responsible for the activity that has been identified. Furthermore, in the discussion, there is supposition that the activity seen by the fraction is due to triterpenoid saponins, but this is based off of previous literature and not the research presented in this paper.
Response: Since putative triterpenoid saponins are main constituents of the fraction, it can be supposed that they are responsible for its biological activity. As I was mentioned in the Discussion, there is no literature information about biological activity of sea buckthorn saponins. Authors of works about saponins purified from sea buckthorn seeds investigated different compounds and they did not report their biological activity. As a result, we had to compare our results with data about saponins from different plant genera.
In the DNA damage assay, it can be seen that DNA damage did not necessarily increase with increasing concentration of the saponin fraction - this is unusual and should be commented on by the authors.
Response: As suggested by the Reviewer, we added the sentence in the description of the results of DNA damage: “Interestingly, DNA damage did not increase with increasing concentration of the saponin fraction”.
Overall, there is merit in this work, although I find that the conclusions that can be made are somewhat limited - while the authors have endeavoured to provide information about the chemical composition of this saponin fraction and also identify biological activity of this fraction, they do not appear to go that step further to attempt to identify what specific components are responsible for the activity seen - this limits the impact of this work.
Response: It must be admitted that it would be much better to investigate activities of purified compounds, it would provide more precise and useful data. Unfortunately, it was not possible at the current stage of our research. Despite this, we believe that our work broadened knowledge about non-phenolic constituents of sea buckthorn leaves and their biological.
